# GXPO: Group Cross-Lingual Relative Policy Optimization for Code Generation

**Linzheng Chai** [1]   **Jian Yang** [1]   **Jiajun Wu** [1]   **Ensheng Shi** [2]   **Xianglong Liu** [1]

## Abstract

Current reinforcement learning (RL) methods for code generation are predominantly optimized on *Python*, showing weak generalization to other programming languages (PLs). Although leveraging multilingual solutions offers richer semantics and a wider search landscape, naive independent training across languages suffers from optimization imbalance and fails to effectively transfer knowledge from high-resource languages. We propose **Group Cross-lingual Relative Policy Optimization (GXPO)**, which forms training groups by generating solutions for the same problem in multiple PLs and jointly optimizes language-specific and cross-language signals, enabling more balanced optimization and improved transfer to low-resource PLs. We additionally introduce **Multilingual LiveCodeBench (ML-LCB)**, extending LiveCodeBench to a unified multilingual evaluation setting. On ML-LCB across 8 PLs, GXPO consistently improves performance, with pronounced gains on low-resource PLs, demonstrating scalable multilingual RL for language-consistent code generation.

## 1. Introduction

Reinforcement learning (RL) has recently shown strong potential in code generation, enabling models to learn executable and test-aware programming behaviors through interaction with compilers and unit tests (Chen et al., 2021; Le et al., 2022; Ouyang et al., 2022). However, current research remains heavily concentrated on Python, with limited exploration of other programming languages (PLs). This monolithic focus restricts the model's generalization capabilities, often resulting in suboptimal performance on languages like C++, Java, and Go. Intuitively, solving a problem across diverse languages creates a wider search

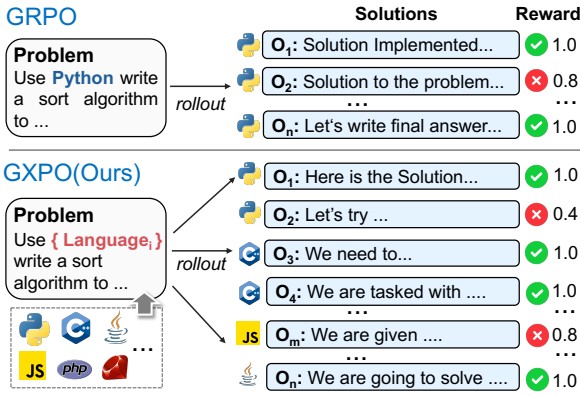

*Figure 1.* Comparison of GRPO and GXPO. Unlike independent sampling, GXPO leverages **multilingual groups** to capture cross-lingual dependencies and facilitate efficient knowledge transfer.

landscape, where distinct syntactic and semantic features can provide complementary feedback, thereby broadening the exploration space during RL training.

Compared to Supervised Fine-Tuning (SFT), RL offers a unique advantage for **multilingual** scaling. While SFT relies on expensive, human-annotated reference solutions for every language, RL only requires language-agnostic test cases (i.e., input-output pairs) to verify correctness. This shared verification environment allows RL to extend coding capabilities from high-resource languages to broader multilingual contexts at minimal data annotation cost.

Despite this potential, naively extending RL to the multilingual setting presents significant challenges. First, training on $L$ languages independently scales computational costs linearly ($L\times$), leading to training inefficiency. Second, varying difficulty levels and intrinsic characteristics of languages often result in optimization imbalance, where the model biases toward languages with denser rewards. Most critically, treating multilingual responses as independent samples fails to exploit the alignment of *parallel corpora*, thereby hindering the effective **transfer of reasoning capabilities** from high-resource languages to low-resource ones.

To address these issues, we propose **GXPO (Group Cross-Lingual Relative Policy Optimization)**, a structured reinforcement learning framework designed for efficient and aligned multilingual code generation. Instead of treating lan-

[1]CCSE, Beihang University [2]Huawei Cloud Computing Technologies. Correspondence to: Jian Yang <jiayang@buaa.edu.cn>.

*Proceedings of the $43^{rd}$ International Conference on Machine Learning*, Seoul, South Korea. PMLR 306, 2026. Copyright 2026 by the author(s).

guages independently, GXPO forms **multilingual groups** for each problem and decomposes the learning signal into three advantages: (1) **Monolingual Advantage**, which stabilizes optimization by comparing responses within the same language; (2) **Cross-lingual Advantage**, which fosters competition by benchmarking a specific language against the group baseline; and (3) **Multilingual Group Advantage**, which explicitly models the marginal contribution of the group's collective success to the overall policy. These mechanism bridges the gap between languages, enabling efficient knowledge transfer and handling reward distribution shifts without merely increasing data volume.

To systematically evaluate this paradigm, we introduce **Multilingual LiveCodeBench (ML-LCB)**, an extension of LiveCodeBench (Jain et al., 2025) that provides a unified evaluation platform across 8 programming languages. Experiments on ML-LCB demonstrate that GXPO significantly outperforms naive baselines, yielding consistent improvements across various base models and delivering particularly substantial gains in low-resource languages, validating the effectiveness of our cooperative optimization strategy.

Our main contributions are summarized as follows:

- We identify critical limitations in naive multilingual RL, including training inefficiency, optimization imbalance, and the failure to leverage parallel semantic structures.

- We propose **GXPO**, a novel framework that integrates Monolingual, Cross-lingual, and Multilingual Group advantages to achieve robust cross-lingual alignment and knowledge transfer.

- We curate **ML-LCB**, the first unified benchmark for multilingual code generation, supporting extensive evaluation across 8 programming languages.

## 2. Related Work

**Code Generation with Large Language Models** The advancement of general-purpose LLMs (OpenAI, 2023; Anthropic, 2024; Meta, 2024; Bai et al., 2023) has led to a specialized line of code-centric models. Early efforts adapted pretrained encoders and decoders for code representation (Feng et al., 2020; Wang et al., 2021; Scao et al., 2022; Yang et al., 2025), while contemporary *code LLMs* emphasize large-scale pretraining and instruction tuning to align with developer intent (Rozière et al., 2023; Guo et al., 2024; Lozhkov et al., 2024; Hui et al., 2024; Huang et al., 2025). Evaluation benchmarks have similarly evolved from basic functional synthesis (Chen et al., 2021; Austin et al., 2021; Zhuo et al., 2024; Wu et al., 2025) to complex scenarios including debugging (Tian et al., 2024; Jimenez et al., 2023; Liu et al., 2025a), multilingual translation (Cassano et al., 2023; Athiwaratkun et al., 2023; Chai et al., 2024), and repository-level reasoning (Liu et al., 2023; Deng et al.,

2024). Notably, LiveCodeBench (Jain et al., 2025) introduces a time-evolving protocol to mitigate data contamination, providing a realistic assessment of competitive programming in python, yet its multilingual potential remains to be fully harnessed.

**Reinforcement Learning with Verifiable Rewards** Reinforcement learning with verifiable rewards (RLVR) has emerged as an effective paradigm for improving reasoning by using deterministic signals as rewards. A significant milestone is GRPO (Shao et al., 2024), which replaces the learned value function with group-based baselines, a strategy later scaled to train state-of-the-art reasoning models like DeepSeek-R1 (Guo et al., 2025). To address optimization challenges such as training instability and length bias, several refined optimizers have been proposed, including methods for decoupling clipping (Yu et al., 2025), mitigating optimization artifacts (Liu et al., 2025b; Cheng et al., 2025), and employing smooth gating (Zheng et al., 2025; Gao et al., 2025). While Zhang et al. (2025) explores multilingual RL for natural language reasoning, code generation remains heavily restricted to Python, leaving cross-lingual transfer underexplored.

## 3. Preliminaries

Generalized Reinforcement Policy Optimization (GRPO) (Shao et al., 2024) builds upon the Proximal Policy Optimization (PPO) framework but eliminates the need for an additional value function by estimating the baseline from group-wise rewards.

Specifically, for each query $q$, GRPO samples a group of $G$ outputs $\{o_1, o_2, \ldots, o_G\}$ from the old policy $\pi_{\theta_{\text{old}}}$. Let $\rho_i(\theta) = \frac{\pi_\theta(o_i|q)}{\pi_{\theta_{\text{old}}}(o_i|q)}$ denote the probability ratio between the current and old policies. The objective function is then simplified as:

$$\mathcal{J}_{\text{GRPO}}(\theta) = \mathbb{E}\left[q \sim P(Q), \{o_i\}_{i=1}^G \sim \pi_{\theta_{\text{old}}}\right]$$

$$\frac{1}{G}\sum_{i=1}^G \Big(\min\Big(\rho_i(\theta)\hat{A}_i, \text{clip}\left(\rho_i(\theta), 1-\varepsilon, 1+\varepsilon\right)\hat{A}_i\Big)$$

$$- \beta\mathbb{D}_{\text{KL}}(\pi_\theta\|\pi_{\text{ref}})\Big)$$

(1)

where $\varepsilon$ and $\beta$ are hyperparameters, and the KL divergence is estimated as:

$$\mathbb{D}_{\text{KL}}(\pi_\theta\|\pi_{\text{ref}}) = \frac{\pi_{\text{ref}}(o_i|q)}{\pi_\theta(o_i|q)} - \log\frac{\pi_{\text{ref}}(o_i|q)}{\pi_\theta(o_i|q)} - 1. \quad (2)$$

The advantage $\hat{A}_i$ is computed by normalizing the rewards $\{R_1, R_2, \ldots, R_G\}$ within the group:

$$\hat{A}_i = \frac{R_i - \text{mean}(\{R_j\}_{j=1}^G)}{\text{std}(\{R_j\}_{j=1}^G)}. \quad (3)$$

# 4. Group Cross-lingual Relative Policy Optimization (GXPO)

## 4.1. Overview

Standard Group Relative Policy Optimization (GRPO) assumes rewards are drawn from a homogeneous distribution. However, in multilingual code generation, different languages exhibit systematic variances in reward statistics due to distinct syntax and execution behaviors. Naively pooling these rewards often biases the policy toward languages with naturally higher pass rates or dilutes fine-grained learning signals through aggressive global normalization.

To address these challenges, we propose **Group Cross-lingual Relative Policy Optimization (GXPO)**. GXPO reformulates the reward normalization process by decomposing the advantage of each response into three complementary dimensions:

- **Monolingual Advantage**: Compares responses within the same language to preserve language-specific variance and internal consistency.
- **Cross-lingual Advantage**: Compares responses across all languages for the same problem, ensuring global competitiveness and cross-lingual alignment.
- **Multilingual Group Advantage**: Evaluates a response's contribution to an idealized multilingual group, capturing cross-language synergy and collective performance.

As shown in Figure 2, GXPO combining these components balances intra-language specialization with inter-language competition, providing a more stable and efficient learning signal for multilingual policy optimization.

## 4.2. Multilingual Rollouts and Rewards

We first describe the rollout and reward computation, corresponding to the top of Figure 2 (*Multilingual Rollout and Reward*). For each training iteration, we sample a mini-batch of questions $\mathcal{Q} = \{q\}$. For each question $q$, we select a set of programming languages $\mathcal{L}_q = \{\ell_1, \dots, \ell_{L_q}\}$ and rollout $G_q^{(\ell)}$ responses for each language $\ell \in \mathcal{L}_q$:

$$o_{q,g}^{(\ell)} \sim \pi_\theta(\cdot \mid q, \ell), \qquad g = 1, \dots, G_q^{(\ell)}. \qquad (4)$$

A verifier executes each program and produces a scalar reward. Let $r_{q,g}^{(\ell)} \in [0,1]$ denote the test pass rate. All subsequent advantages are computed from $\{r_{q,g}^{(\ell)}\}$.

## 4.3. Monolingual, Cross-lingual, and Multilingual Advantages

GXPO defines three advantages for each response, reflecting three different views of performance. All views are standardized using only statistics from the current mini-batch, so no moving averages are required.

**Monolingual Advantage**   The monolingual advantage measures how good a response is compared to *other responses in the same language*. we follow Hu et al. (2025), aggregated over all questions in the batch. For each language $\ell$, we collect:

$$\mathcal{R}^{(\ell)} = \big\{ r_{q,g}^{(\ell)} \mid q \in \mathcal{Q}, \ g = 1, \dots, G_q^{(\ell)} \big\}, \qquad (5)$$

and compute the batch-level mean and variance:

$$\mu_\ell^{\mathrm{mono}} = \frac{1}{|\mathcal{R}^{(\ell)}|} \sum_{q,g} r_{q,g}^{(\ell)}, \qquad (6)$$

$$\sigma_\ell^{\mathrm{mono}} = \sqrt{\frac{1}{|\mathcal{R}^{(\ell)}|} \sum_{q,g} \big(r_{q,g}^{(\ell)} - \mu_\ell^{\mathrm{mono}}\big)^2}. \qquad (7)$$

To avoid numerical instability when a language appears only a few times, we clamp the standard deviation with a small constant $\tau_{\mathrm{mono}} > 0$.

The monolingual advantage for a sample $(q, \ell, g)$ is:

$$A_{q,g}^{\mathrm{mono},(\ell)} = \frac{r_{q,g}^{(\ell)} - \mu_\ell^{\mathrm{mono}}}{\max(\sigma_\ell^{\mathrm{mono}}, \tau_{\mathrm{mono}})}. \qquad (8)$$

This term corresponds to the *Monolingual Advantage* block in Figure 2.

**Cross-lingual Advantage**   The cross-lingual advantage compares a response with all other responses (across languages) for the *same* question $q$. For each $q$ we collect

$$\mathcal{R}_q = \big\{ r_{q,g}^{(\ell)} \mid \ell \in \mathcal{L}_q, \ g = 1, \dots, G_q^{(\ell)} \big\},$$

and compute the question-level statistics:

$$\mu_q^{\mathrm{cross}} = \frac{1}{|\mathcal{R}_q|} \sum_{\ell,g} r_{q,g}^{(\ell)}, \qquad (9)$$

$$\sigma_q^{\mathrm{cross}} = \sqrt{\frac{1}{|\mathcal{R}_q|} \sum_{\ell,g} \big(r_{q,g}^{(\ell)} - \mu_q^{\mathrm{cross}}\big)^2}. \qquad (10)$$

The cross-lingual advantage for $(q, \ell, g)$ is:

$$A_{q,g}^{\mathrm{cross},(\ell)} = \frac{r_{q,g}^{(\ell)} - \mu_q^{\mathrm{cross}}}{\max(\sigma_q^{\mathrm{cross}}, \tau_{\mathrm{cross}})}, \qquad (11)$$

where $\tau_{\mathrm{cross}} > 0$ is a small constant. This term forms the *Cross-lingual Advantage* block in Figure 2, highlighting whether a particular response is better or worse than the average multilingual performance on the same question.

**Multilingual Group Advantage**   Finally, we define a multilingual group advantage that matches the *Multilingual Rollout* and *Multilingual Groups* part of Figure 2. Conceptually, we consider an idealized process where a multilingual

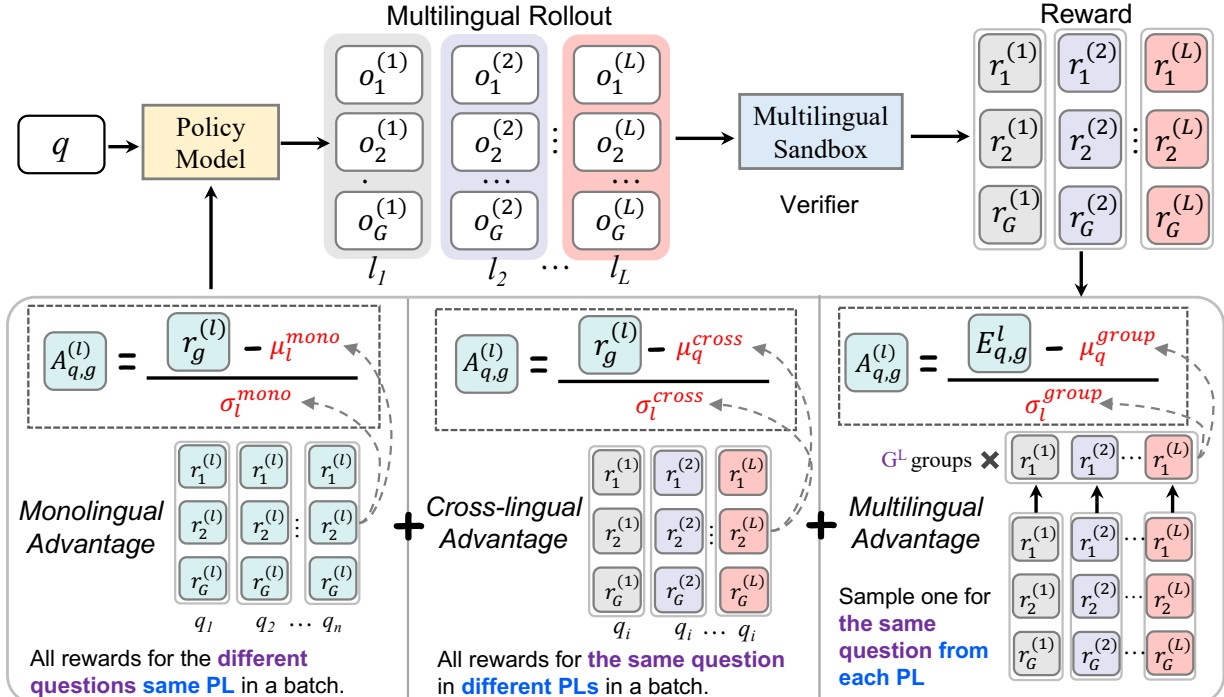

*Figure 2.* **Overview of the GXPO.** For a given question $q$, the policy model generates multiple responses across different programming languages (PLs). These responses are evaluated in a sandbox to obtain rewards. The final training signal is a combination of three advantage components: (i) **Monolingual Advantage**, which normalizes rewards within the same PL across the entire batch; (ii) **Cross-lingual Advantage**, which normalizes rewards across all PLs for a specific question; and (iii) **Multilingual Advantage**, which calculates the expected contribution of a response to a multilingual group formed by sampling one candidate from each PL. (For brevity, we omit the question subscript $q$ in reward terms $r$ and the advantage-type superscripts (mono/cross/multi) in $A$.)

group is formed by selecting *one* response from each language and scoring the group by averaging the rewards of its members.

For each question $q$ and language $\ell$, we first compute the per-language mean reward on that question:

$$\bar{r}_q^{(\ell)} = \frac{1}{G_q^{(\ell)}} \sum_{g=1}^{G_q^{(\ell)}} r_{q,g}^{(\ell)}. \qquad (12)$$

Let $L_q = |\mathcal{L}_q|$ denote the number of languages used for $q$.

Now consider a random multilingual group constructed by independently sampling one response from each language, uniformly over the observed candidates. Conditioned on choosing a specific response $(q, \ell, g)$ for language $\ell$, the *expected group reward* is:

$$E_{q,g}^{\text{group},(\ell)} = \frac{1}{L_q} \left( r_{q,g}^{(\ell)} + \sum_{\substack{m \in \mathcal{L}_q \\ m \neq \ell}} \bar{r}_q^{(m)} \right). \qquad (13)$$

This expectation aggregates the actual reward of the chosen response with the average behavior of other languages on the same question, and can be computed analytically from the

existing rollouts without explicitly enumerating all language combinations.

We then compute question-level statistics over these expected group rewards:

$$\mu_q^{\text{group}} = \frac{1}{N_q} \sum_{\ell \in \mathcal{L}_q} \sum_{g=1}^{G_q^{(\ell)}} E_{q,g}^{\text{group},(\ell)}, \qquad (14)$$

$$\sigma_q^{\text{group}} = \sqrt{\frac{1}{N_q} \sum_{\ell,g} \left( E_{q,g}^{\text{group},(\ell)} - \mu_q^{\text{group}} \right)^2}, \qquad (15)$$

where $N_q = \sum_\ell G_q^{(\ell)}$ is the total number of responses for question $q$.

The multilingual group advantage for $(q, \ell, g)$ is:

$$A_{q,g}^{\text{multi},(\ell)} = \frac{E_{q,g}^{\text{group},(\ell)} - \mu_q^{\text{group}}}{\max(\sigma_q^{\text{group}}, \tau_{\text{group}})}, \qquad (16)$$

with a small $\tau_{\text{group}} > 0$ to stabilize variance.

This term explicitly measures how well a response performs when it participates in a multilingual team, capturing *multilingual synergy.*

### 4.4. Combined GXPO Advantage and Policy Optimization

For each response $(q, \ell, g)$ we combine the three views into a single GXPO advantage by a weighted sum:

$$A_{q,g}^{\text{GXPO},(\ell)} = \lambda_1 A_{q,g}^{\text{mono},(\ell)} + \lambda_2 A_{q,g}^{\text{cross},(\ell)} + \lambda_3 A_{q,g}^{\text{multi},(\ell)}, \quad (17)$$

where $\lambda_1, \lambda_2, \lambda_3 \geq 0$ are hyper-parameters controlling the relative importance of monolingual learning, cross-lingual competition, and multilingual cooperation.

Given these sequence-level advantages, we broadcast them to tokens using the response mask and apply a standard GRPO-style objective. Let $\rho_{q,g}^{(\ell)}$ denote the ratio between the new and old policy probabilities of the sampled trajectory (sequence-level likelihood ratio). The GXPO loss is:

$$\mathcal{L}_{\text{GXPO}} = -\mathbb{E}_{q,\ell,g} \left[ \min \left( \rho_{q,g}^{(\ell)} A_{q,g}^{(\text{GXPO},\ell)}, \right. \right.$$

$$\left. \left. \text{clip}(\rho_{q,g}^{(\ell)}, 1-\epsilon, 1+\epsilon) A_{q,g}^{(\text{GXPO},\ell)} \right) \right] \quad (18)$$

optionally augmented with KL regularization to a reference policy and an entropy bonus, as in standard PPO.

Algorithm 1 summarizes the full training loop with GXPO advantages. GXPO extends GRPO from single-language rollouts to *multilingual cooperative optimization*. By decomposing the advantage into a monolingual component (language-wise consistency), a cross-lingual component (within-problem competition), and a multilingual group component (expected group performance under joint multilingual rollouts), GXPO produces stable, well-aligned learning signals across languages and improves sample efficiency for multilingual code generation.

## 5. Experiments

### 5.1. Dataset

**Training Dataset**   For reinforcement learning, we construct a high-quality training dataset based on CodeContest-Plus (Wang et al., 2025b), an open-source collection of competitive programming problems. The original dataset comprises approximately 10K problems, each featuring a natural-language description, input/output test cases, and a set of user submissions. To guarantee the reliability of rewards during RL training, we filter the dataset by evaluating the True Positive Rate (TPR) and True Negative Rate (TNR) of solutions against their test cases. This ensures that the test suites are of high quality and can accurately distinguish between valid and invalid code. After filtering, we obtain 5,335 high-quality problems. Since the dataset exclusively uses the Standard I/O (Stdio) format, it is easily extensible to various programming languages. We incorporate 8 diverse languages, namely Python, Java, C++, JavaScript, Go, PHP,

---

**Algorithm 1** Overview of the GXPO

**Require:** Base policy $\pi_{\theta^{(0)}}$, training steps $T$, language set $\mathcal{L}$, per-language rollout counts $\{G^{(\ell)}\}$.
1: **for** $t = 1, \ldots, T$ **do**
2:    Sample a mini-batch of questions $\mathcal{Q} = \{q\}$.
3:    **for** each $q \in \mathcal{Q}$ and each language $\ell \in \mathcal{L}_q$ **do**
4:       Roll out $G_q^{(\ell)}$ responses $\{o_{q,g}^{(\ell)}\}_{g=1}^{G_q^{(\ell)}} \sim \pi_{\theta^{(t-1)}}(\cdot \mid q, \ell)$.
5:       Execute programs with the verifier and compute scalar rewards $r_{q,g}^{(\ell)}$.
6:    **end for**
7:    Compute monolingual statistics $\mu_\ell^{\text{mono}}, \sigma_\ell^{\text{mono}}$ for all $\ell$ and obtain $A_{q,g}^{\text{mono},(\ell)}$.
8:    For each question $q$, compute $\mu_q^{\text{cross}}, \sigma_q^{\text{cross}}$ and obtain $A_{q,g}^{\text{cross},(\ell)}$.
9:    For each question $q$ and language $\ell$, compute $\bar{r}_q^{(\ell)}$, then $E_{q,g}^{\text{group},(\ell)}$, and $A_{q,g}^{\text{multi},(\ell)}$.
10:    Combine the three advantages into $A_{q,g}^{\text{GXPO},(\ell)}$.
11:    Broadcast sequence-level advantages to tokens, compute the GXPO loss $\mathcal{L}_{\text{GXPO}}$, and update $\theta^{(t)}$ by gradient descent.
12: **end for**

---

Ruby, and Perl, which are selected for their widespread popularity and their representation of different programming paradigms. This enables us to perform multilingual training and rollouts under a unified evaluation framework.

**Evaluation Benchmark**   For evaluation, we develop Multi-Language LiveCodeBench (ML-LCB), an extension of LiveCodeBench-v5 and v6 (Jain et al., 2025), which contain 167 and 175 problems, respectively. By utilizing periodically updated problems, ML-LCB mitigates data leakage and ensures a robust assessment of model coding capabilities. Following the methodology of Wang et al. (2025a), we convert all problems into the standard I/O format. This approach bypasses the complexities of language-specific function signatures and declarations, ensuring a more unified execution environment. Consistent with our training set, ML-LCB supports code generation and automatic judging across the same 8 programming languages.

### 5.2. Experiment Setup

We implement GXPO on the *verl* framework (Sheng et al., 2024) for distributed training. For each problem, the model generates 4 candidate responses across 8 programming languages, resulting in 32 multilingual samples per group. To ensure a fair comparison, the rollout size for the GRPO baseline is also set to 32. The policy update batch size is 16, with a clipping ratio $\epsilon = 0.2$. The KL divergence penalty is omitted to further accelerate the training process.

The reward $r_i \in [0, 1]$ is defined as the unit test pass rate. Optimization utilizes the GXPO loss with weighting coefficients $(\lambda_1, \lambda_2, \lambda_3)$ empirically set to $(0.6, 0.2, 0.2)$. Training is performed for 600 steps with a global batch size of 32

*Table 1.* **Per-language Pass@1 (%, higher is better)** on *Multilingual LiveCodeBench* (ML-LCB). **Top:** v5 results. **Bottom:** v6 results. High PLs include Python, Java, CPP, and JS; Low PLs include Go, PHP, Ruby, and Perl.

| Method | Python | Java | CPP | JS | Go | PHP | Ruby | Perl | High | Low | **Avg.** |
|---|---|---|---|---|---|---|---|---|---|---|---|
| **ML-LCB v5** | | | | | | | | | | | |
| *Qwen3-4B-Base* | | | | | | | | | | | |
| GRPO | 17.4 | 9.6 | 12.0 | 11.4 | **13.2** | 13.8 | 12.0 | 11.4 | 12.6 | 12.6 | 12.6 |
| **GXPO** | **17.4**+0.0 | **15.6**+6.0 | **13.2**+1.2 | **16.8**+5.4 | 12.6-0.6 | **13.8**+0.0 | **12.6**+0.6 | **13.2**+1.8 | **15.8**+3.2 | **13.1**+0.5 | **14.4**+1.8 |
| *Qwen2.5-Coder-7B-Instruct* | | | | | | | | | | | |
| w/o Training | 13.2 | 9.0 | 7.2 | 2.4 | 6.6 | 8.4 | 10.8 | 9.0 | 8.0 | 8.7 | 8.3 |
| GRPO | 18.0 | 15.6 | **14.4** | 18.6 | 16.2 | 19.2 | 14.4 | 16.8 | 16.7 | 16.7 | 16.6 |
| **GXPO** | **23.4**+5.4 | **16.8**+1.2 | 13.8-0.6 | **19.2**+0.6 | **16.2**+0.0 | **24.6**+5.4 | **18.6**+4.2 | **18.6**+1.8 | **18.3**+1.6 | **19.5**+2.8 | **18.9**+2.3 |
| *LLaMA-3.1-8B* | | | | | | | | | | | |
| w/o Training | 15.6 | 10.8 | 9.0 | 5.4 | 6.6 | 8.4 | 10.8 | 5.4 | 10.2 | 7.8 | 9.0 |
| GRPO | 14.4 | 12.0 | **15.6** | 12.0 | 12.0 | 15.0 | 12.0 | 11.4 | 13.5 | 12.6 | 13.0 |
| **GXPO** | **19.8**+5.4 | **14.4**+2.4 | 12.0-3.6 | **12.6**+0.6 | **16.8**+4.8 | **16.2**+1.2 | **18.0**+6.0 | **13.2**+1.8 | **14.7**+1.2 | **16.1**+3.5 | **15.4**+2.4 |
| **ML-LCB v6** | | | | | | | | | | | |
| *Qwen3-4B-Base* | | | | | | | | | | | |
| GRPO | **19.4** | 13.7 | 16.0 | 14.9 | 11.4 | 14.3 | **18.3** | 12.0 | 16.0 | 14.0 | 15.0 |
| **GXPO** | 18.9-0.5 | **18.9**+5.2 | **21.1**+5.1 | **21.1**+6.2 | **18.3**+6.9 | **21.1**+6.8 | 16.6-1.7 | **17.1**+5.1 | **20.0**+4.0 | **18.3**+4.3 | **19.1**+4.1 |
| *Qwen2.5-Coder-7B-Instruct* | | | | | | | | | | | |
| w/o Training | 16.0 | 12.0 | 15.4 | 2.3 | 6.3 | 12.6 | 14.3 | 12.0 | 11.4 | 11.3 | 11.3 |
| GRPO | **24.6** | 21.1 | 19.4 | 24.0 | 17.1 | 21.7 | 24.0 | 22.3 | 22.3 | 21.3 | 21.8 |
| **GXPO** | 22.9-1.7 | **22.3**+1.2 | **20.0**+0.6 | **24.0**+0.0 | **20.6**+3.5 | **24.0**+2.3 | **24.6**+0.6 | **22.3**+0.0 | **22.3**+0.0 | **22.9**+1.6 | **22.6**+0.8 |
| *LLaMA-3.1-8B* | | | | | | | | | | | |
| w/o Training | 15.4 | 12.0 | 9.7 | 2.3 | 3.4 | 13.7 | 10.9 | 6.3 | 9.9 | 8.6 | 9.2 |
| GRPO | 16.0 | 12.0 | 9.7 | 14.3 | 12.6 | 15.4 | 12.6 | 0.0 | 13.0 | 10.2 | 11.6 |
| **GXPO** | **17.1**+1.1 | **14.2**+2.2 | **13.1**+3.4 | **14.9**+0.6 | **12.6**+0.0 | **15.4**+0.0 | **16.6**+4.0 | **14.4**+14.4 | **14.8**+1.8 | **14.8**+4.6 | **14.8**+3.2 |

and a maximum output sequence length of 8,192 tokens. All experiments are conducted on $16 \times$ NVIDIA H20 (96GB) GPUs. Models performance are evaluated using the Pass@1 (averaged over 4 runs) metric (Chen et al., 2021), with the temperature set to 0.6 and $top\_p$ set to 0.95.

We evaluate GXPO using three strong open-source models: **Qwen3-4B-Base** (Qwen, 2025), **Qwen2.5-Coder-7B** (Hui et al., 2024), and **LLaMA-3.1-8B** (Meta, 2024). We compared GXPO with the scores before model training (excluding the base model) and the GRPO method.

# 6. Results

## 6.1. Overall Performance on ML-LCB

Table 1 summarizes the Pass@1 results, where GXPO consistently outperforms GRPO across all evaluated models and benchmark versions. Specifically, on ML-LCB v6, GXPO delivers average improvements of **+4.1%** for Qwen3-4B-Base and **+3.2%** for LLaMA-3.1-8B compared to the GRPO baseline. To better understand the performance distribution, we categorize the eight programming languages into two groups: High PLs (Python, Java, CPP, JS), which have abundant training resources, and Low PLs (Go, PHP, Ruby, Perl), which are relatively underrepresented. When examining the performance gap between these two groups,

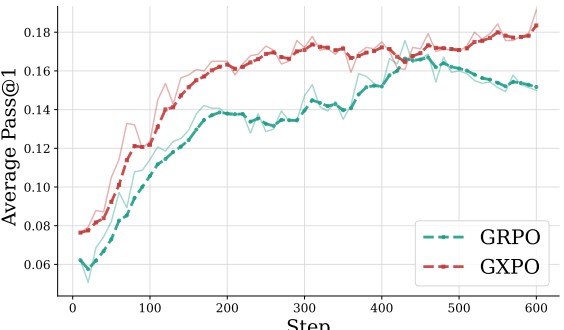

*Figure 3.* Average Pass@1 on ML-LCB v6 during training process. GXPO demonstrates superior learning efficiency and higher final convergence compared to the GRPO baseline.

GXPO demonstrates more balanced results; for example, on LLaMA-3.1-8B (v6), GXPO achieves comparable scores across both groups (High: 14.8%, Low: 14.8%), whereas GRPO shows a noticeable gap (High: 13.0%, Low: 10.2%). Notably, we observe that GRPO exhibits instability in language balancing; for instance, in the LLaMA-3.1-8B (v6) setup, GRPO's performance on Perl dropped to **0.0%**, suggesting that standard reinforcement learning may struggle to maintain performance across diverse languages simultaneously. In contrast, GXPO achieves **14.4%** in the same setting, indicating that its group-level multilingual optimization helps mitigate language interference and provides more

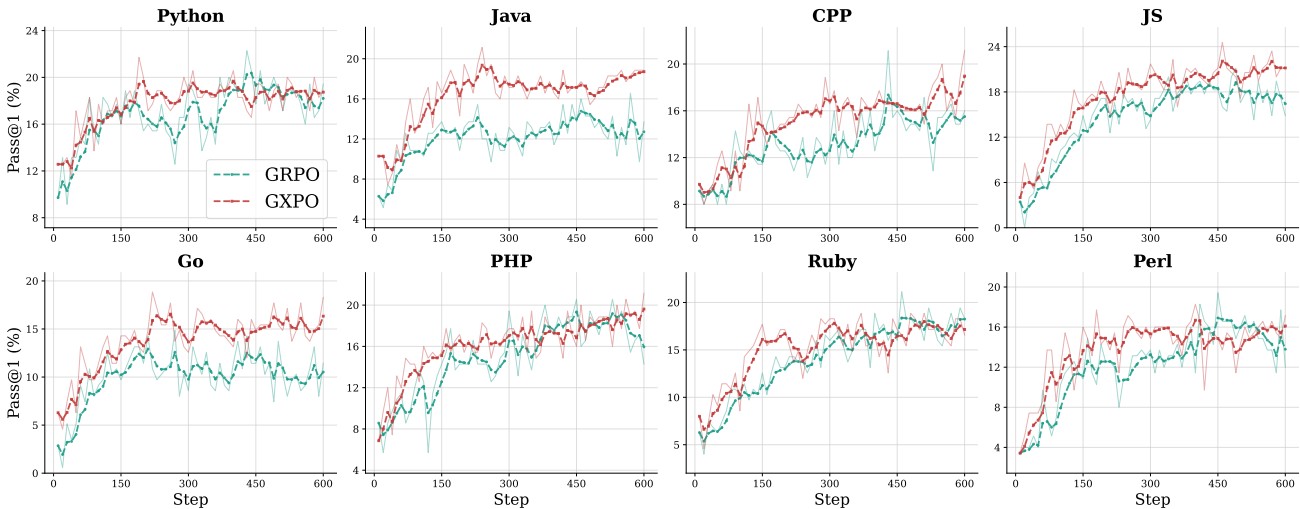

*Figure 4.* Per-language training dynamics on ML-LCB v6. GXPO consistently maintains higher performance and better stability across diverse languages, effectively mitigating the training fluctuations observed in GRPO.

stable training signals for multilingual code generation.

### 6.2. Performance during Training

To analyze the training dynamics of GXPO, we monitor the performance on ML-LCB v6 (Qwen3-4B-Base) throughout the reinforcement learning process. Figure 3 illustrates the average and per-language performance during training.

**Learning Efficiency and Stability.** In Figure 3, GXPO exhibits a more favorable learning trajectory compared to GRPO. While both methods start from the same baseline, GXPO shows an early advantage and maintains a steady upward trend. GRPO's performance begins to plateau and fluctuate in later stages, whereas GXPO reaches a relatively higher and more stable convergence. This suggests that GXPO provides more consistent gradient updates by balancing reward signals across different language distributions.

**Cross-lingual Robustness.** The per-language breakdown (Figure 4) illustrates the behavior of GXPO in handling language interference. In languages such as Java, CPP, and Go, GXPO tends to outperform GRPO with lower variance. In contrast, GRPO exhibits noticeable fluctuations, particularly in Perl and Go, which aligns with our observation in Table 1. These results suggest that GXPO's group-level optimization contributes to more balanced performance across multiple programming languages.

## 7. Further Analysis

### 7.1. Ablation Studies

As shown in Table 2, each component of GXPO contributes distinctively to multilingual code generation. Removing the monolingual advantage (*w/o Mono Adv.*) leads to the largest

*Table 2.* **Ablations on GXPO components**. We report overall Avg pass@1 on ML-LCB v6. "w/o Mono Adv." removes the monolingual advantage $A^{\mathrm{mono}}$ ; "w/o Cross Adv." removes the cross-lingual advantage $A^{\mathrm{cross}}$; "w/o Multi Adv." removes the multilingual group advantage $A^{\mathrm{multi}}$.

| Method | Avg pass@1 | High Avg. | Low Avg. |
|---|---|---|---|
| GRPO | 21.8 | 22.3 | 21.3 |
| GXPO | **22.6** | **22.3** | **22.9** |
|   w/o Mono Adv. | 20.9 | 21.3 | 20.4 |
|   w/o Cross Adv. | 21.6 | 21.3 | 22.0 |
|   w/o Multi Adv. | 21.1 | 21.8 | 20.4 |

drop (from 22.6% to 20.9%), particularly affecting Low-resource PLs, suggesting its role in mitigating high-resource bias through language-specific normalization. Excluding the cross-lingual advantage (*w/o Cross Adv.*) results in a moderate decline to 21.6%, indicating the benefit of within-question cross-language comparison. Removing the multilingual group advantage (*w/o Multi Adv.*) also decreases performance, especially for Low-resource PLs, highlighting the contribution of expected group rewards to cross-language synergy. These results suggest that all three components contribute positively to multilingual code generation.

### 7.2. Cross-Lingual Correlation Transfer Analysis

In Figure 6, we characterize cross-language coupling by tracking per-language pass@1 at each training checkpoint and computing an $8 \times 8$ Pearson correlation matrix over the resulting score trajectories (one trajectory per language). To reveal structure, correlations are converted to distances $(1 - r)$ and average-linkage agglomerative clustering is applied; the matrix is then reordered by the clustering leaf order (computed independently for each method) to produce the clustered heatmaps.

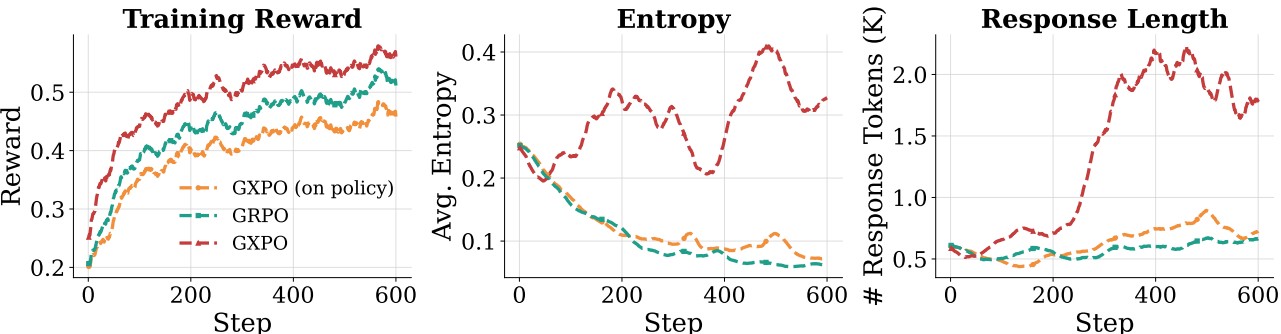

*Figure 5.* **Comparison of training dynamics between GXPO and GRPO baselines.** We monitor (left) training reward, (middle) response length in tokens, and (right) average policy entropy. GXPO (red) demonstrates higher reward ceilings, a characteristic surge in response length indicating exploratory reasoning, and a significantly more stable entropy profile compared to the GRPO baseline.

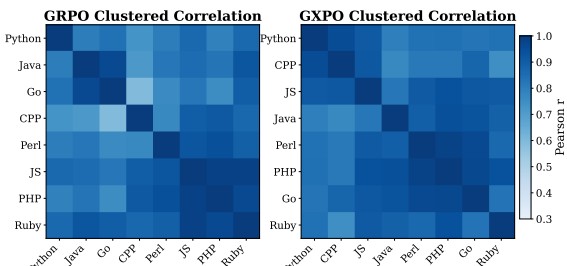

*Figure 6.* Language–language Pearson correlation of pass@1 scores across training steps for GRPO (left) and GXPO (right). Darker blue indicates stronger positive correlation.

The clustered heatmaps indicate consistently high positive correlations across languages, implying that performance changes tend to co-move rather than trade off. Under the shared color scale, GXPO shows darker and more uniform correlations than GRPO, suggesting stronger and more stable cross-language synchronization. The block-like patterns further suggest that some languages form tighter co-varying groups; these blocks appear less fragmented for GXPO, consistent with more coordinated multilingual progress.

### 7.3. RL Training Process

To gain a deeper understanding of the optimization process, we monitor the training reward, response length, and policy entropy throughout the reinforcement learning phase. In addition to the default GXPO setting, we also evaluate an on-policy variant, denoted as GXPO (on-policy), which increases the policy update batch size from 16 to 32, resulting in fewer gradient updates per training iteration. As illustrated in Figure 5, GXPO exhibits distinct characteristics compared to the GRPO baseline.

**Training Reward.** As shown in the left panel of Figure 5, all methods exhibit a steady upward trend in reward, indicating successful policy optimization. **GXPO** consistently maintains a higher reward compared to GRPO and its on-policy variant, with the gap becoming more pronounced

as training progresses. The GXPO(on-policy) shows lower performance due to fewer policy updates.

**Policy Entropy.** The middle panel highlights a notable distinction in training dynamics. While GRPO and the on-policy variant maintain relatively low entropy levels (around 0.1), **GXPO** exhibits higher and more dynamic entropy throughout training, with values fluctuating between 0.2 and 0.4. This suggests that GXPO promotes more exploratory behavior and maintains greater output diversity, potentially helping the model avoid premature convergence to suboptimal solutions.

**Response Length.** An interesting phenomenon is observed in the right panel: while the response lengths of GRPO and GXPO (on-policy) remain relatively stable below 1,000 tokens, **GXPO** exhibits a more rapid growth in response length, peaking at over 2,000 tokens around step 400–500. This increase suggests that GXPO encourages the model to allocate more computational budget for reasoning, thereby expanding the exploration space and enhancing its ability to tackle complex problems across diverse programming languages.

### 8. Conclusion

In this work, we introduced Group Cross-lingual Relative Policy Optimization (GXPO), a novel reinforcement learning framework that enhances multilingual code generation by decomposing learning signals into monolingual, cross-lingual, and group-level advantages. By leveraging semantically equivalent problems across diverse programming languages, GXPO effectively addresses optimization imbalance and facilitates efficient knowledge transfer from high-resource to low-resource languages. Extensive evaluations on our curated Multilingual LiveCodeBench (ML-LCB) across 8 programming languages demonstrate that GXPO consistently outperforms baseline methods, providing a robust and scalable solution for achieving language-consistent performance in code generation.

# Acknowledgements

This work is supported by the Fundamental Research Funds for the Central Universitie (Grant No.GW2025-19) and supported by State Key Laboratory of Complex & Critical Software Environment(Grant No. SKLCCSE-2025ZX-26).

# Impact Statement

This paper presents work whose goal is to advance the field of machine learning. There are many potential societal consequences of our work, none of which we feel must be specifically highlighted here.

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
