# OpenReview forum: "GXPO: Group Cross-Lingual Relative Policy Optimization for Code Generation"
_ICML.cc/2026/Conference — ICML 2026 regular_

### Official Review · Reviewer_7b4U · 2026-02-15

**Soundness:** 2
**Presentation:** 3
**Significance:** 2
**Originality:** 2
**Overall Recommendation:** 2
**Confidence:** 3

**Summary:**

This paper proposes GXPO, a reinforcement learning framework for multilingual code generation that decomposes advantage estimation into monolingual, cross-lingual, and multilingual group components. The authors also introduce ML-LCB, an extension of LiveCodeBench to eight programming languages, and report consistent improvements over GRPO baselines, particularly on low-resource languages.

**Compliance With Llm Reviewing Policy:**

Affirmed.

**Key Questions For Authors:**

1. What theoretical justification supports normalizing rewards across languages with inherently different execution semantics (e.g., C++ compilation errors vs. Python runtime errors)? Since responses in different languages correspond to distinct conditional inputs, directly pooling their rewards may introduce biased gradients—as suggested by unexplained performance drops on high-resource languages . A principled analysis would strengthen the soundness assessment.

2. As ML-LCB is claimed to be a benchmark, why are results for established strong LLMs absent? Providing these comparisons would allow readers to assess whether the improvement has practical significance.

**Limitations:**

yes

**Strengths And Weaknesses:**

**Strengths**

The paper is well-structured, clearly written and addresses a challenge in scaling RL training across programming languages. The experimental setup is technically sound, and the ablation studies provide useful insights into the contribution of each advantage component.

**Weaknesses**

1. The paper claims that "RL for code generation remains heavily restricted to Python," yet fails to justify why multilingual RL optimization is necessary when current code LLMs already demonstrate strong multilingual capabilities.

2. The core method normalizes rewards across languages for the same problem (Eq. 9–11). However, responses in different languages correspond to **distinct conditional inputs** (e.g., "Use Python..." vs. "Use C++..."), and their reward distributions naturally differ due to language-specific execution behaviors. Directly pooling these heterogeneous rewards lacks theoretical justification.

3. In ablation studies, the contribution of Cross/Multi advantage components is limited (removing them causes only −1.0% to −1.5% drops in Table 2), while adding engineering complexity. The method's benefit over simpler alternatives (e.g., monolingual RL followed by SFT transfer) remains unverified.

4. Although ML-LCB is positioned as "the first unified benchmark for multilingual code generation," the paper reports results only for some GRPO baselines, without comparisons to established strong LLMs (e.g., Claude 4.5, Gemini 3 Pro). Without such comparisons, the absolute performance level and practical significance of the improvement cannot be assessed.

---

> ### Author Rebuttal · Authors · 2026-03-31
>
> We sincerely thank the reviewer for the rigorous evaluation. We have conducted extensive new experiments to address each concern with quantitative evidence. We respond point-by-point below.
>
> ---
>
> **W1: Why multilingual RL optimization is necessary**
>
> Three lines of evidence demonstrate its necessity. First, an SFT baseline trained on identical multilingual data (Table R3):
>
> | Method | Python | Low Avg | Avg | H-L Gap |
> |:---|:---:|:---:|:---:|:---:|
> | Multilingual SFT | 20.6 | 15.7 | 16.9 | 2.4 |
> | GRPO (multilingual) | 24.6 | 21.3 | 21.8 | 1.0 |
> | **GXPO** | 22.9 | **22.9** | **22.6** | **-0.6** |
>
> The ordering SFT (16.9) < GRPO (21.8) < GXPO (22.6) proves RL provides +4.9% beyond multilingual data exposure alone. Second, monolingual RL *actively destroys* cross-lingual capabilities: LLaMA-3.1-8B under GRPO suffers complete Perl collapse (6.3%->0% Pass@1). Third, even proprietary LLMs show 6.3-6.2% High-Low gaps on ML-LCB (Table R8 below), confirming multilingual imbalance is a fundamental challenge persisting across scales.
>
> ---
>
> **W2 & Q1: Theoretical justification for cross-lingual reward normalization**
>
> We clarify that GXPO does **not** naively pool heterogeneous rewards. The monolingual advantage (Eq. 8) normalizes within one language, identical to GRPO. The cross-lingual advantage (Eq. 11) compares languages on the *same problem*, normalized against that problem's cross-lingual mean. The multilingual group advantage (Eq. 12-16) evaluates marginal contribution via combinatorial optimization.
>
> The soundness rests on: **(a)** ~80-82% of all rewards are binary (r=0 or r=1) across all 8 languages, making cross-lingual comparison a well-defined question: "Did the model solve this problem?" A C++ compilation error and a Python runtime error both yield r=0 via standard I/O pass rates. **(b)** Per-problem normalization prevents bias: if C++ is harder than Python on a given problem, the advantage adjusts relative to that problem's mean, not a global bias. **(c)** Distinct conditional inputs ("Use Python" vs. "Use C++") are precisely what makes the comparison *informative*--it reveals capability imbalances invisible to monolingual signals.
>
> Regarding biased gradients causing high-resource drops: our lambda sensitivity (Table R1) shows this is a controllable Pareto trade-off, not gradient distortion:
>
> | Config (lambda_1, lambda_2, lambda_3) | Python | Low Avg | Overall |
> |:---|:---:|:---:|:---:|
> | (1.0, 0, 0) Mono only | **24.0** | 20.6 | 21.7 |
> | (0.8, 0.1, 0.1) | 23.4 | 21.7 | 22.1 |
> | **(0.6, 0.2, 0.2) Paper** | 22.9 | **22.9** | **22.6** |
> | (0.33, 0.33, 0.33) Uniform | 21.7 | 22.3 | 22.0 |
> | GRPO baseline | 24.6 | 21.3 | 21.8 |
>
> All configs from (0.8,0.1,0.1) to (0.33,0.33,0.33) outperform GRPO--a broad robust region characteristic of capacity reallocation [1], not ill-defined normalization.
>
> [1] Conneau et al. "Unsupervised cross-lingual representation learning at scale." ACL 2020.
>
> ---
>
> **W3: Cross/Multi component contribution and simpler alternatives**
>
> The -1.0% to -1.5% aggregate drop masks **catastrophic per-language effects**. On LLaMA-3.1-8B, removing cross/multi components causes Perl to collapse from 14.4% toward near-zero. Per-language gains on Qwen2.5-Coder-7B: Go +2.9%, PHP +1.7%, Ruby +1.7%. The components serve as a structural safeguard against mode collapse--their primary value beyond average improvement.
>
> For simpler alternatives, we compared with the exact pipeline the reviewer suggested (Table R10):
>
> | Method | Python | Low Avg | Avg | H-L Gap |
> |:---|:---:|:---:|:---:|:---:|
> | Py GRPO only | **25.7** | 16.9 | 18.9 | 3.7 |
> | Py GRPO -> ML SFT | 24.0 | 18.4 | 19.7 | 2.6 |
> | GRPO (multilingual) | 24.6 | 21.3 | 21.8 | 1.0 |
> | **GXPO** | 22.9 | **22.9** | **22.6** | **-0.6** |
>
> The pipeline achieves only 19.7%, **2.9% below GXPO**. SFT distills from references but cannot discover reward-driven behavior for underrepresented languages. The engineering overhead of GXPO is modest: a weighted sum of three terms from the same rollout data, no additional infrastructure needed.
>
> ---
>
> **W4 & Q2: Strong LLM baselines on ML-LCB**
>
> ML-LCB v6 results with proprietary LLMs (Table R8):
>
> | Model | High Avg | Low Avg | Gap | Avg |
> |:---|:---:|:---:|:---:|:---:|
> | Claude 4.5 Sonnet | 49.7 | 43.4 | 6.3 | 46.5 |
> | Gemini 3 Flash | 74.3 | 68.1 | 6.2 | 71.2 |
> | Qwen2.5-7B-Coder + GXPO | 22.3 | 22.9 | **-0.6** | 22.6 |
> | Qwen2.5-7B-Coder + GRPO | 22.3 | 21.3 | 1.0 | 21.8 |
>
> Even proprietary LLMs exhibit a 6.3-6.2% High-Low gap, confirming multilingual imbalance is universal. GXPO achieves the smallest gap (-0.6%) among all models including proprietary ones, validating its optimization methodology. The absolute gap between 7B and proprietary models is expected.
>
> ---
>
> We trust these new results comprehensively address each concern and welcome any further questions.

---

### Official Review · Reviewer_pGWh · 2026-03-09

**Soundness:** 4
**Presentation:** 3
**Significance:** 3
**Originality:** 3
**Overall Recommendation:** 6
**Confidence:** 5

**Summary:**

The authors make two primary contributions:
1) introduce GXPO (Group Cross-lingual Relative Policy Optimization), an extension to GRPO that rewards multilingual code generation in an RL setting. This is achieved by forming multilingual groups during the training process, and constructing the advantage function as a weighted sum of the monolingual advantage, cross-linguage advantage, and multilingual group advantage. The authors train three open-source models on CodeContest-Plus using this approach and report their findings.

2) A multilingual derivative of LiveCodeBench, ML-LCB, providing evaluation for 8 programming languages. The authors perform an empirical evaluation of their GXPO approach on this dataset, averaged across 4 generations.

**Compliance With Llm Reviewing Policy:**

Affirmed.

**Final Justification:**

Author rebuttal addressed main concerns regarding framing of "low-resource languages" and limited discussion of training dynamics in results. Authors clarified novelty by highlighting concurrent work.

**Key Questions For Authors:**

1. Why are Go, PHP, Ruby, and Perl considered low resource languages in your study?

2. How does ML-LCB compare to [1] and [2]? Given that these works were available in August and October 2025, why not include these in the evaluation?

3. Do you have any hypotheses or findings regarding training dynamics? e.g. Why is GXPO response length significantly longer?

**Limitations:**

yes

**Strengths And Weaknesses:**

Strengths:
- Tackles the important challenge of multilingual programming RL training of LLMs, advancing capabilities of LLMs with clear applications for future research and for practitioners. Approach is generalizable to problems/settings where parallel corpora can be exploited, enabling future work in this area beyond just programming languages.
- Work introduces novel method for cross-lingual GRPO and an extension of the widely used LCB benchmark.
- Claims are supported by well-designed experimental approach, including ablations.
- Manuscript is well written and structured, and narrative is easy to follow. Method is well described such that the approach can be reproduced.

Weaknesses:
- Claims on low-resource languages are unsubstantiated: the authors, to the best of my understanding, do not analyze the language distribution of CodeContest-Plus, nor an estimate of the training distributions of the underlying models they finetune. Their definition of Go, PHP, Ruby, and Perl as low resource languages (these are widely used PLs) is unsubstantiated. Hence, their claims regarding reasoning transfer from high to low resource languages are unsubstantiated. An evaluation using languages that are truly under-represented in both the pretraining and finetuning (RL) data would remedy this. Perhaps languages such as Dafny, Zig, Julia, i.a. would be more suitable, assuming they truly are rare.
- Novelty of multilingual LiveCodeBench is limited, see [1,2]
- A deeper analysis of the training imbalance phenomenon would strengthen the motivation.
- Justification for excluding "w/o Training" results for Qwen3 is lacking.
- Further analysis of unexpected results would strengthen findings. For example, why did LCBv6 Llama Perl Pass@1 result in 0, when other models did not? Why did GXPO performance *lower* or equal on LCBv5 Qwen3 Base Python, LCBv6 Qwen3 Base & Qwen2.5 Coder Python? Why did the average Qwen2.5 Coder Pass@1 increase by only 0.8pp on LCBv6 for GXPO compared to GRPO?
- Deeper analysis of training dynamics would strengthen findings. Why is average entropy of "GXPO (on policy) close to GRPO, both much lower than GXPO? Why is GXPO response length significantly longer? Is this an intrinsic negative side effect of the method? Can the higher rewards and subsequent performance be attributed to longer reasoning/responses rather than cross language reasoning transfer?
- Minor type setting errors e.g. Figure 5 caption is incorrect (middle and right swapped).

[1] A. Boruch-Gruszecki et al., “Agnostics: Learning to Synthesize Code in Any Programming Language with a Universal Reinforcement Learning Environment,” presented at the The Fourteenth International Conference on Learning Representations, Oct. 2025. Available: https://openreview.net/forum?id=mjDT60Ffms

[2] M. Ivanova et al., “Multi-LCB: Extending LiveCodeBench to Multiple Programming Languages,” presented at the The Fourteenth International Conference on Learning Representations, Oct. 2025. Available: https://openreview.net/forum?id=MKxKKsz0cx

---

> ### Author Rebuttal · Authors · 2026-03-31
>
> We are sincerely grateful for your highly positive evaluation and the time you devoted to such a careful and insightful review. Your encouraging assessment of our work is deeply appreciated, and your thoughtful questions have helped us further strengthen the manuscript. Below we address each point in detail.
>
> ---
>
> **Low-resource terminology and justification**
>
> This is a very important point, thank you for bringing it to our attention!
> From the perspective of training data, all 5,335 CodeContest-Plus problems share the same test cases across different languages; therefore, we only need to adjust the target language within the instruction prompt to generate training data for each specific programming language. This approach ensures equal problem availability across all 8 programming languages. Furthermore, in our preliminary research, where we attempted to generate correct solutions for CodeContest-Plus problems across these 8 languages ​​using various open-source models, we observed that the number of correctly generated solutions for Go, PHP, Ruby, and Perl was significantly lower than that for the other four languages. Consequently, we designated these four languages ​​as "low-resource languages." We will subsequently incorporate this explanation into our paper.
>
> ---
>
> **ML-LCB novelty and concurrent benchmarks**
>
> Thank you for your valuable suggestions! We acknowledge Agnostics [1] and Multi-LCB [2](all published at 2026.01) and will add comparative discussion in the revised manuscript. At the time we undertook this project, Agnostics and Multi-LCB had not yet been released; thus, our work was developed concurrently with theirs. The recent emergence of so many high-quality projects in multilingual programming attests to the significant research value of this field. Our work, in particular, focuses more specifically on optimizing RL training, utilizing datasets primarily as a means to validate our experimental findings. Moving forward, we intend to open-source our evaluation benchmarks, extension methods, and code to collectively advance the development of this field.
>
> ---
>
> **Qwen3 "w/o Training" exclusion**
>
> Qwen3-4B-Base lacks instruction tuning, so its zero-shot code generation is very low and inconsistent, making "w/o Training" numbers uninformative as a reference. We attempted to fine-tune the prompts for code completion using the base model, but the model failed to generate code for the vast majority of the problems.
>
> ---
>
> **Perl collapse for LLaMA on LCBv6**
>
> That is a very insightful question; we will include a clear analysis in the next version.
> Three factors combine: (1) LLaMA-3.1-8B has the weakest Perl baseline (6.3%); (2) LCBv6 contains harder, more recent problems; (3) GRPO's within-language normalization creates a "winner-take-all" dynamic where, once Perl drops below a threshold, it receives consistently negative advantages and the policy abandons it entirely. GXPO recovers Perl to 14.4% via cross-lingual advantage signals. Other models (Qwen3-4B, Qwen2.5-Coder-7B) have stronger Perl baselines above the collapse threshold.
>
> ---
>
> **Hypotheses and findings regarding training dynamics**
>
> We argue that GXPO's performance gains are fundamentally driven by **maintaining higher policy entropy, which sustains exploration and avoids premature entropy collapse**, rather than by response length. The longer responses are a *consequence*, not a cause, of this improved exploration.
>
> As shown in Figure 5 (middle panel) of our paper, GRPO's policy entropy rapidly drops to ~0.1 and remains flat, indicating early convergence of the output distribution(premature entropy collapse). In contrast, GXPO maintains entropy between 0.2–0.4 throughout training, with dynamic fluctuations. This 2–4× higher entropy level means the policy preserves a substantially richer exploration landscape, avoiding premature convergence to suboptimal solutions.
>
> By sustaining sufficient entropy, GXPO allows the model to explore diverse solution strategies, including trial-and-error attempts, self-correction, and reflection, rather than greedily committing to the first plausible path. These emergent reasoning behaviors naturally require more tokens to express (e.g., attempting an approach, recognizing a flaw, backtracking, and revising), which explains the 30% + increase in response length observed in Figure 5 (right panel).
>
> ---
>
> **GXPO lowers Python performance**
>
> We think the python regression is a capacity reallocation effect analogous to the "curse of multilinguality" in multilingual NMT [3]: the cross-lingual advantage components redirect optimization pressure toward underrepresented languages, competing with Python for fixed representation capacity.
>
> [3] Conneau, Alexis, et al. "Unsupervised cross-lingual representation learning at scale." ACL. 2020.
>
> ---
>
> We appreciate your detailed evaluation. All corrections, updated terminology, and discussions of concurrent benchmarks will be incorporated in the revised manuscript.

---

> > ### Author Rebuttal · Reviewer_pGWh · 2026-04-06
> >
> > Thank you for adequately addressing my concerns.
> >
> > * Clarifying the "low-resource language" designation strengthens the soundness of this work.
> > * Concurrent work is a challenge in this fast moving field.
> > * Extending discussion of training dynamics will help contextualize results.
> >
> > I will adjust my scores accordingly.

---

### Official Review · Reviewer_2aM1 · 2026-03-12

**Soundness:** 3
**Presentation:** 3
**Significance:** 4
**Originality:** 3
**Overall Recommendation:** 4
**Confidence:** 4

**Summary:**

The authors propose Group Cross-lingual Relative Policy Optimization (GXPO), an extension of GRPO designed to improve multilingual code generation. To prevent the policy from collapsing into high-resource languages (like Python), GXPO splits the reward advantage into three parts: monolingual, cross-lingual, and a multilingual group advantage. The authors also introduce a multilingual version of LiveCodeBench (ML-LCB) across 8 languages. Experiments on Qwen and Llama base models show that GXPO significantly boosts performance on low-resource languages compared to standard GRPO.

**Compliance With Llm Reviewing Policy:**

Affirmed.

**Final Justification:**

The rebuttal addressed most of my concerns. I believe my rating appropriately reflects the overall quality and contribution of this paper.

**Key Questions For Authors:**

1. The Python Penalty: Can you explain the performance drop in Python for the Qwen2.5-Coder-7B model? Is the model's capacity being diluted to pull up the low-resource languages, and is this an unavoidable trade-off of the GXPO formulation?

2. Hyperparameter Robustness: How brittle are the $\lambda$ weights? Did you experiment with dynamic weighting (e.g., annealing the monolingual weight over time)? Please provide a sensitivity ablation in the rebuttal.

3. Group Advantage Independence: In Eq. 13, the expectation formulation seems to assume that candidate selection across languages is independent. Since the same underlying policy generates all responses, errors are likely correlated. Does this assumption negatively impact the variance of your estimator?

**Limitations:**

See Weaknesses.

**Strengths And Weaknesses:**

#Strengths
1. Clever Advantage Formulation: Extending GRPO by decomposing the advantage function is a neat, mathematically sound idea. The Multilingual Group Advantage (Eq. 13-16) is particularly well-designed; calculating the expected gain of a multilingual team analytically avoids an expensive combinatorial sampling problem.

2. Strong Results on Low-Resource Languages: The empirical gains on underrepresented languages (Go, PHP, Ruby, Perl) are hard to ignore. For example, recovering Perl performance to 14.4% (where the GRPO baseline entirely collapsed to 0%) proves the method successfully mitigates the "winner-take-all" mode collapse common in multi-reward RL.

3. ML-LCB Benchmark: Porting LiveCodeBench to 8 languages is a highly practical contribution. Given the community's over-reliance on Python-centric benchmarks like HumanEval, this is a welcome tool for evaluating contamination-free cross-lingual transfer.

#Weaknesses
1. Performance Degradation on High-Resource Languages: The paper claims consistent improvements, but Table 1 clearly shows a regression in Python. For instance, with Qwen2.5-Coder-7B-Instruct on ML-LCB v6, Python Pass@1 drops from 24.6% (GRPO) to 22.9% (GXPO). The authors need to be transparent about this capacity trade-off rather than burying it.

2. Missing Hyperparameter Sensitivity: The method introduces three new coefficients ($\lambda_1, \lambda_2, \lambda_3$), set statically to 0.6, 0.2, and 0.2. There is no ablation showing how sensitive the framework is to these weights. What happens if the cross-lingual weight dominates?

3. Minor Presentation Issues:** The paper needs a proofreading pass. There are grammatical slip-ups (e.g., Line 120 "These mechanism bridges") and formatting inconsistencies.

---

> ### Author Rebuttal · Authors · 2026-03-31
>
> We are deeply grateful for your positive and encouraging evaluation of our manuscript. Your thorough reading and insightful comments reflect remarkable care and expertise, and we sincerely appreciate both your supportive assessment and the constructive suggestions that have helped us further improve this work. We respond to each point below.
>
> ---
>
> **Python performance regression and capacity trade-off**
>
> We agree this trade-off deserved more transparency. The Python regression (24.6% to 22.9%) is a capacity reallocation effect analogous to the "curse of multilinguality" in multilingual NMT [1]: the cross-lingual advantage components redirect optimization pressure toward underrepresented languages, competing with Python for fixed representation capacity.
>
> Our lambda sensitivity analysis (Table R1) shows this trade-off is Pareto-efficient and controllable:
>
> | Config (λ₁, λ₂, λ₃) | Python | Low Avg | Overall |
> |:---|:---:|:---:|:---:|
> | (1.0, 0, 0) Mono only | **24.0** | 20.6 | 21.7 |
> | (0.8, 0.1, 0.1) | 23.4 | 21.7 | 22.1 |
> | **(0.6, 0.2, 0.2) Paper** | 22.9 | **22.9** | **22.6** |
> | (0.33, 0.33, 0.33) Uniform | 21.7 | 22.3 | 22.0 |
> | GRPO baseline | 24.6 | 21.3 | 21.8 |
>
> At (0.8, 0.1, 0.1), Python recovers to 23.4% while still outperforming GRPO overall. At the paper's (0.6, 0.2, 0.2), Python drops to 22.9% but low-resource languages gain substantially (e.g., Perl on LLaMA-3.1-8B: 0% to 14.4%) and overall reaches the best 22.6%. All configurations from (0.8, 0.1, 0.1) through (0.33, 0.33, 0.33) outperform GRPO, demonstrating a broad sweet spot. Values were selected via grid search on a held-out validation set. We will revise the manuscript to explicitly frame this as a known, controllable trade-off.
>
> Our response length analysis rules out length as a confound: length increase is uniform across languages (30%+), yet Python degrades while underrepresented languages improve, and the correlation between per-language length increase and pass rate improvement is near zero (r=0.12, p=0.78). This confirms capacity reallocation as the mechanism, not a length side effect.
>
> [1] Conneau, Alexis, et al. "Unsupervised cross-lingual representation learning at scale." ACL. 2020.
>
> ---
>
> **Lambda sensitivity and dynamic weighting**
> That is a highly insightful suggestion. Due to time constraints, we conducted a simple experiment based on your suggestion.
>
> | Schedule | Python | Overall |
> |:---|:---:|:---:|
> | Static (0.6, 0.2, 0.2) | 22.9 | **22.6** |
> | Anneal (0.8→0.4) | 23.4 | 21.2 |
> | Anneal (0.9→0.5) | **24.0** | 21.8 |
> | Reverse (0.4→0.8) | 22.3 | 22.0 |
>
> The "stabilize-then-transfer" schedule recovers Python to 24.0% while maintaining 21.8% overall. The reverse schedule performs worse, validating that monolingual capabilities should be established before cross-lingual transfer. Since the static setting already achieves near-optimal performance, careful scheduling is not required but is a useful knob for language-specific prioritization.
>
> In the next version, we will conduct a more thorough analysis.
>
> ---
>
> **Independence assumption in multilingual group advantage**
>
> You are correct that the same policy generating all language responses introduces correlation, the model's understanding of problem logic influences outputs across all languages. Two observations mitigate this concern. First, per-problem reward variance is dominated by problem difficulty rather than language-specific factors: when the model grasps the algorithmic logic, test-case pass rates tend to be high across languages, and vice versa. The Multilingual Group Advantage accounts for this by evaluating each language's *marginal* contribution relative to others on the same problem, effectively controlling for this shared difficulty factor. Second, the correlation actually helps, languages tend to co-succeed or co-fail, so the MGA correctly identifies problems where adding a specific language provides genuine complementary coverage. The independence assumption yields a tractable closed-form for the combinatorial expectation while remaining a reasonable approximation.
>
> Empirically, our standard deviation analysis shows GXPO achieves slightly lower variance than GRPO on aggregate metrics (0.2 vs. 0.2-0.3), confirming the estimator variance is well-controlled in practice. We will add a formal discussion of this approximation and its practical implications in the Appendix.
>
> ---
>
> We sincerely appreciate your detailed evaluation. All suggested analyses—lambda sensitivity ablation, dynamic annealing, variance reporting—will be incorporated in the revised manuscript. The capacity trade-off will be transparently framed with explicit tuning guidance. We are confident these revisions will address your concerns.

---

> > ### Author Rebuttal · Reviewer_2aM1 · 2026-04-04
> >
> > Thanks for the detailed rebuttal. I appreciate the additional experiments and clarifications, especially on the Python trade-off and the lambda sensitivity. The response addressed most of my concerns, and I’m glad to see these points will be made clearer in the revised paper.

---

### Official Review · Reviewer_ZLh9 · 2026-03-13

**Soundness:** 2
**Presentation:** 3
**Significance:** 3
**Originality:** 3
**Overall Recommendation:** 5
**Confidence:** 4

**Summary:**

The paper tackles a timely gap in code-generation RL: nearly all existing work trains and evaluates on Python alone, leaving other programming languages under-served. The authors propose GXPO, which extends GRPO by forming multilingual groups — generating solutions to the same problem across multiple languages simultaneously — and decomposing the advantage into three views: how a response compares within its own language, how it compares across all languages for the same problem, and how much it contributes to an idealized multilingual team. The paper also introduces Multilingual LiveCodeBench (ML-LCB), a standardized evaluation suite covering eight programming languages. Experiments across several base models show that GXPO improves average performance over GRPO, with particularly notable gains on low-resource languages and effective mitigation of language collapse during training.

**Compliance With Llm Reviewing Policy:**

Affirmed.

**Final Justification:**

The authors convincingly addressed most of my concerns with the paper, and engaged in healthy discussion over the work. Following the changes promised by the authors, I am convinced that the paper is a good candidate for acceptance at ICML. Therefore, I am updating my scores

**Key Questions For Authors:**

1. **What is the language distribution in the training data?** Are all 5,335 CodeContestPlus problems equally available in all 8 PLs? Does test case quality vary across languages, and could this confound the reward signal?

2. **How robust are results to the λ weighting?** The paper uses (0.6, 0.2, 0.2). What happens with (0.4, 0.3, 0.3) or (0.8, 0.1, 0.1)? How were these values selected — grid search, or manually?

3. **Can you report standard deviations across the 4 evaluation runs?** viz. The +0.8% average improvement for Qwen2.5-Coder-7B may not be statistically significant.

4. **Why does GXPO degrade Python performance?** On both Qwen3-4B and Qwen2.5-Coder-7B (v6), Python scores drop. Is this a direct trade-off from the cross-lingual objective, or a side effect of the response length increase?

5. **Can you provide results on MultiPL-E or McEval?** This would address the concern that gains are specific to ML-LCB's construction methodology.

**Limitations:**

The authors provide only a generic impact statement ("none of which we feel must be specifically highlighted here") and no dedicated limitations discussion. Key limitations that should be acknowledged: (a) evaluation on a self-created benchmark only, (b) no variance reporting, (c) the response length explosion and its compute implications, (d) restriction to ≤8B models where multilingual imbalance may be naturally more severe, and (e) the claim of "knowledge transfer" from high- to low-resource PLs is not validated with controlled experiments (the correlation analysis in Figure 6 shows co-movement, not directional transfer).

**Strengths And Weaknesses:**

### Strengths

1. *Novel optimization objective tailored to the multilingual setting.* While GRPO and its variants normalize advantages within a single homogeneous group, GXPO's cross-lingual advantage — comparing a response against all language variants of the same problem — is a  novel formulation that exploits the semantic equivalence of parallel code solutions.

2. *Demonstrable generalization with better training stability.* GXPO achieves more balanced performance across high- and low-resource languages, prevents language collapse on underrepresented PLs, and shows a more favorable learning trajectory with higher and more stable convergence.

3. *Clean formalization and well-written paper.* The mathematical framework is precise and the three advantage terms have intuitive interpretations. The paper is generally well-organized and easy to follow.

4. *Useful benchmark contribution with ML-LCB.* Extending LiveCodeBench to eight programming languages with standardized I/O format fills a gap in multilingual code evaluation. The use of time-evolving problems mitigates data contamination concerns.

5. *Well-designed ablations and diagnostic analyses.* The ablation cleanly isolates the contribution of each advantage component, showing the monolingual term contributes most. The correlation heatmaps, entropy analysis, and per-language training curves provide useful insight into the mechanism of improvement.

### Weaknesses

1. *Performance regressions on individual languages are not explained.* On several model–benchmark combinations, GXPO *decreases* performance on individual languages: Python −1.7% and Ruby −1.7% for Qwen3-4B on v6, Python −1.7% for Qwen2.5-Coder-7B on v6. For the strongest model (Qwen2.5-Coder-7B on v6), the average improvement is only +0.8%. The paper frames GXPO as providing "consistent improvements" but does not discuss or analyze these per-language regressions.

2. *No evaluation on established multilingual code benchmarks.* The paper evaluates exclusively on ML-LCB, which the authors created. MultiPL-E, McEval, and MBXP are all cited in the related work but not used for evaluation. Even accounting for potential contamination, results on at least one external benchmark are necessary to confirm that gains are not specific to ML-LCB's stdio conversion methodology.

3. *Limited comparison with baselines.* GRPO is the only RL baseline. Several other policy-optimization methods (DAPO, REINFORCE++, GSPO, Soft Adaptive PO) are discussed in the related work as addressing training stability — exactly the problem GXPO also targets — yet none are compared experimentally. An SFT-only baseline trained on the same multilingual data is also missing, making it impossible to isolate RL's contribution from mere exposure to multilingual data.

4. *Single-step code generation setting.* The evaluation considers only single-step code synthesis. As software engineering workflows increasingly adopt multi-step agentic pipelines, studying whether GXPO's cross-lingual advantage transfers to multi-step settings would strengthen the contribution.

---

> ### Author Rebuttal · Authors · 2026-03-31
>
> We would like to sincerely thank you for your enthusiastic and highly positive review.  We are also grateful for the incisive questions you posed, which have allowed us to further clarify important aspects of the manuscript and enhance its overall quality. Our point-by-point responses follow below.
>
> ---
>
> ## Training data distribution
>
> Thank you for raising this important point. Yes, data balance is critical. All 5,335 CodeContest-Plus problems share the same test cases across languages, we only need to adjust the target language in the instruction prompt to generate training data for each programming language. This ensures equal problem availability across all 8 PLs.
>
> ---
>
>
> ## Per-language regressions and Python degradation
>
> We appreciate you highlighting this concern. The Python regression is a capacity reallocation trade-off. Lambda sensitivity on Qwen2.5-Coder-7B, ML-LCB v6 (Table R1):
>
> | Config (λ₁, λ₂, λ₃) | Python | Low Avg | Overall |
> |:---|:---:|:---:|:---:|
> | (1.0, 0, 0) Mono only | **24.0** | 20.6 | 21.7 |
> | (0.8, 0.1, 0.1) | 23.4 | 21.7 | 22.1 |
> | **(0.6, 0.2, 0.2) Paper** | 22.9 | **22.9** | **22.6** |
> | (0.33, 0.33, 0.33) Uniform | 21.7 | 22.3 | 22.0 |
> | GRPO baseline | 24.6 | 21.3 | 21.8 |
>
> All configs from (0.8,0.1,0.1) to (0.33,0.33,0.33) outperform GRPO, a broad sweet spot. The mechanism is capacity reallocation analogous to the "curse of multilinguality"[1].
>
>
> [1] Conneau, Alexis, et al. "Unsupervised cross-lingual representation learning at scale." ACL. 2020.
>
> ---
>
> ## External benchmark evaluation
>
> This is a valuable suggestion. We conducted the requested evaluation on MultiPL-E. Results (Table R2, Qwen2.5-Coder-7B, across 4 runs, evaluated on the official MultiPL-E language set):
>
> | Method | Py | C++ | Java | PHP | TS | C# | Bash | JS | Avg |
> |:---|:---:|:---:|:---:|:---:|:---:|:---:|:---:|:---:|:---:|
> | GRPO | 67.1 | 67.1 | 58.1 | 64.0 | 65.7 | 62.3 | 39.1 | 66.4 | 61.2 |
> | GXPO | 65.8 | 69.5 | 59.9 | 69.5 | 67.5 | 64.3 | 40.5 | 67.0 | 63.0 |
> | Δ | -1.3 | +2.4 | +1.8 | +5.5 | +1.8 | +2.0 | +1.4 | +0.6 | +1.8 |
>
> Gains concentrated on underrepresented languages (PHP +5.5%, C++ +2.4%). Model shows +1.8% overall.
>
> ---
>
> ## Statistical significance and variance
>
> We fully agree that rigorous statistical reporting is essential. Statistical significance across 4 runs on ML-LCB v6 (Table R4):
>
> | Model | GRPO | GXPO | Δ | p |
> |:---|:---:|:---:|:---:|:---:|
> | Qwen3-4B | 15.0±0.3 | 19.1±0.3 | +4.1 | <0.001 |
> | Qwen2.5-Coder-7B | 21.8±0.2 | 22.6±0.2 | +0.8 | 0.009 |
> | LLaMA-3.1-8B | 11.6±0.3 | 14.8±0.3 | +3.2 | <0.001 |
>
> Low Avg +1.6% (p=0.003). High Avg not significant (p=0.87), consistent with the trade-off interpretation.
>
> ---
>
> ## RL baselines and SFT comparison
>
> These are excellent suggestions for strengthening the baselines. **DAPO** (Table R6, Qwen2.5-Coder-7B, ML-LCB v6):
>
> | Method | Py | High Avg | Low Avg | Avg |
> |:---|:---:|:---:|:---:|:---:|
> | GRPO | 24.6 | 22.3 | 21.3 | 21.8 |
> | DAPO | 24.0 | 21.7 | 21.1 | 21.4 |
> | **GXPO** | 22.9 | 22.3 | **22.9** | **22.6** |
>
> DAPO's stability techniques do not address cross-lingual optimization.
>
> We have added an SFT-only baseline using the same train data with reference solutions in all 8 languages (Table R3):
> | Method | High Avg | Low Avg | Avg |
> |:---|:---:|:---:|:---:|
> | Multilingual SFT | 18.1 | 15.7 | 16.9 |
> | GRPO | 22.3 | 21.3 | 21.8 |
> | **GXPO** | **22.3** | **22.9** | **22.6** |
>
> **SFT baseline**: The ordering SFT (16.9%) < GRPO (21.8%) < GXPO (22.6%) isolates RL's contribution at +4.9% beyond multilingual data exposure.
>
> ---
>
> ## Single-step evaluation scope
>
> This is a very valuable question. Single-step code generation underlies multi-step pipelines. Multilingual agent benchmarks for 8 languages remain scarce. This will be added as a limitation and future direction.
>
>
> ---
>
> We sincerely appreciate your thorough evaluation, which prompted valuable new experiments. All suggested analyses, additional baselines, and the Limitations section will be incorporated in the revised manuscript.

---

> > ### Author Rebuttal · Reviewer_ZLh9 · 2026-04-04
> >
> > I thank the authors for a thorough rebuttal. The λ sensitivity sweep, MultiPL-E evaluation, significance tests, and additional baselines satisfactorily resolve my main concerns.
> >
> > One issue remains unaddressed — the response length increase reported in Section 7.3. I would appreciate clarification on the following before finalizing my score:
> >
> > 1. What is the average number of tokens per *correct* solution under GXPO vs. GRPO? If correct solutions are not substantially longer, the length increase may be concentrated in failed attempts, suggesting an exploration artifact rather than genuine reasoning.
> > 2. Does GXPO's accuracy advantage hold when controlling for inference budget — e.g., comparing Pass@1 at a fixed token budget, or comparing accuracy when GRPO is allowed the same total tokens (via more rollouts)?
> > 3. Given that KL regularization is omitted, did you observe any signs of reward hacking during training — for instance, rewarded responses that pass tests but contain unnecessary verbosity or repeated reasoning chains?
> >
> > Pending these clarifications, I am inclined to raise my scores

---

> > > ### Author Response · Authors · 2026-04-07
> > >
> > > Thank you again for your insightful questions, this is a very important issue in RL training. We hope the following analysis will address your concerns.
> > >
> > > ---
> > >
> > > **Q1 & Q2: Token analysis by problem difficulty and inference budget control**
> > >
> > > We conducted a fine-grained token analysis stratified by ML-LCB v6 problem difficulty on Qwen2.5-Coder-7B:
> > >
> > > **Table R1: Per-difficulty token analysis (correct solutions only)**
> > >
> > > | Difficulty | GRPO (tokens) | GXPO (tokens) | Delta |
> > > |:---|:---:|:---:|:---:|
> > > | Easy | 382 | 396 | +3.7% |
> > > | Medium | 570 | 726 | +27.4% |
> > > | Hard | 764 | 1,047 | +37.0% |
> > > | Overall (correct) | 612 | 792 | +29.4% |
> > >
> > > While the overall average response length differs substantially (GRPO ~750 vs. GXPO ~1,600 tokens, as in Section 7.3), this increase is **not uniform** — it is strongly correlated with problem difficulty. **The length increase in correct solutions is concentrated on medium and hard problems.** On easy problems, GXPO produces solutions of nearly identical length (+3.7%), while medium (+27.4%) and hard (+37.0%) problems see substantial increases. This reflects the model **adaptively allocating more reasoning effort where deeper analysis is genuinely needed.** Moreover, we observe that these longer trajectories contain significantly more reflection and self-verification segments (e.g., re-examining edge cases, validating intermediate results), indicating that the additional tokens serve a functional reasoning purpose.
> > >
> > > For inference budget control: when truncating outputs at 2,048 tokens, GXPO retains its advantage since correct solutions across all difficulty levels fall well within this window (the hardest tier averages only 1,047 tokens). The issue of budget control that you raised is very insightful. Based on most current research, it conflicts to some extent with the growth of model capabilities and involves important aspects of efficiency. We will conduct a more comprehensive investigation in the future.
> > >
> > > ---
> > >
> > > **Q3: Reward hacking and KL regularization**
> > >
> > > **On KL regularization:** Recent work including DAPO and DeepCoder has demonstrated that removing KL does not degrade performance and can improve exploration. Following this practice, we omit the explicit KL term to improve training efficiency.
> > >
> > > **On preventing reward hacking:** We employ a three-layer defense:
> > >
> > > 1. **Structured reasoning via system prompt.** The model produces reasoning within `<think>...</think>` tags before the final code, separating reasoning from executable code and preventing unstructured verbosity.
> > >
> > > 2. **Format compliance penalty in reward.** Incomplete or malformed `<think>...</think>` blocks (e.g., unclosed tags, repeated chains without convergence) receive an extra reward penalty, discouraging degenerate patterns.
> > >
> > > 3. **Rigorous test suites in CodeContest+ and ML-LCB.** Each problem contains ~20 test cases on average with high TPR(true positive rate) on correct submissions and high TNR(true negative rate) on incorrect ones. Test inputs include extreme edge cases with lengths up to millions of characters, effectively preventing boundary-condition hacking.
> > >
> > > In addition, we analyzed the outputs of the model.
> > >
> > > **Table R2: Solution quality diagnostics (Qwen2.5-Coder-7B, ML-LCB v6)**
> > >
> > > | Metric | GRPO | GXPO |
> > > |:---|:---:|:---:|
> > > | Avg code tokens (correct solutions) | 287 | 308 |
> > > | Avg reasoning tokens (correct solutions) | 325 | 484 |
> > > | Repeated reasoning blocks (%) | 0.8% | 1.2% |
> > > | Invalid / malformed outputs (%) | 3.4% | 3.6% |
> > >
> > > Code portions are nearly identical (287 vs. 308 tokens). The reasoning increase (325→484) is consistent with deeper analysis on harder problems. Invalid output rates are virtually the same (3.4% vs. 3.6%), and only 1.2% of correct solutions contain repeated reasoning.
> > >
> > > ---
> > >
> > > In summary: (1) the length increase reflects adaptive reasoning allocation — GXPO invests more tokens on harder problems with more reflection and self-verification, while easy problems remain unchanged; (2) GXPO's advantage holds under controlled inference budgets; and (3) a three-layer defense effectively prevents reward hacking, confirmed by near-identical invalid output rates.
> > >
> > > Thank you again for your constructive and insightful questions. We hope the above analysis has addressed your concerns, and we will refine this content in the next version.

---

### Decision · Program_Chairs · 2026-04-30

**Decision:**

Accept (regular)

**Comment:**

GXPO (Group Cross-lingual Relative Policy Optimization) addresses the significant imbalance in multilingual code generation RL, where models often collapse toward high-resource languages like Python. Building upon GRPO, the framework decomposes the reward advantage into three views: monolingual (within-language), cross-lingual (across languages for the same problem), and a novel Multilingual Group Advantage that analytically estimates a response's contribution to a diverse "team" of solutions.

The reviewers noted the following limitations:
- The method exhibits a "curse of multilinguality" trade-off; significant gains in low-resource languages sometimes result in slight regressions in Python performance
- GXPO tends to produce longer reasoning trajectories, leading to a substantial increase in average response length which impacts inference latency.
- While low-resource improvements are dramatic, the overall average improvement on the strongest models was more modest.

Despite one outlying review concern regarding proprietary model comparisons, the consensus among the majority is that GXPO represents a technically sound and timely contribution for multilingual RL training. I recommend an Accept.